# Apricot (*Prunus armeniaca*) Performance under Foliar Application of Humic Acid, Brassinosteroids, and Seaweed Extract

Adel M. Al-Saif [1,*], Lidia Sas-Paszt [2], Rehab M. Awad [3] and Walid F. A. Mosa [3]

1   Department of Plant Production, College of Food and Agriculture Sciences, King Saud University, P.O. Box 2460, Riyadh 11451, Saudi Arabia
2   The National Institute of Horticultural Research, Konstytucji 3 Maja 1/3, 96-100 Skierniewice, Poland
3   Plant Production Department (Horticulture-Pomology), Faculty of Agriculture, Saba Basha, Alexandria University, Alexandria 21531, Egypt
*   Correspondence: adelsaif@ksu.edu.sa

**Abstract:** The excessive use of chemical fertilizers in fruit orchards has led to numerous problems for the environment, produce quality, and food safety. It also negatively affects soil health, beneficial microorganisms, and ground water quality, hence the resurgence of the application of biostimulants as ecofriendly ways to improve the growth, yield, and fruit quality of tree fruits. The current study was performed during 2021 and 2022 to investigate the influence of foliar spraying of 500, 1000, and 2000 mg/L humic acid (HA); 0.5, 1, and 2 mg/L brassinosteroids (Brs); and 1000, 2000, and 3000 mg/L seaweed extract (SWE) compared with a control (untreated trees) in terms of the performance of an apricot (*Prunus armeniaca*) cv. Canino. The obtained results show that the spraying of HA, Brs, and SWE positively increased the shoot length, leaf area, leaf chlorophyll content, fruit set, fruit yields, and fruit physical and chemical characteristics, as well as leaf macro- or micronutrients contents compared with those untreated trees during both study years. Moreover, the increase in parameter values was parallel to the increase in the used concentrations of HA, Brs, or SWE, where 2000 mg/L HA, 2 mg/L Brs, and 3000 mg/L SWE were superior to 1000 mg/L HA, 1 mg/L Brs, and 2000 mg/L SWE, which were better than 500 mg/L HA, 0.5 mg/L Brs, and 1000 mg/L SWE.

**Keywords:** biostimulants; fruit quality; *Fruit yields*; leaf nutritional status; *Prunus armeniaca*

## 1. Introduction

Apricot (*Prunus armeniaca*) are rich in sugars; proteins; crude fiber; crude fat; total minerals; vitamins such as an A, C, K, and B complex; and organic acids such as citric and malic acids. In addition, apricot are characterized by high contents of total phenolic compounds, flavonoids, polysaccharides, polyphenol, fatty acid, sterol derivatives, carotenoids, cyanogenic glycosides, and volatile components, so they can be used to treat asthma, cough, constipation, chronic gastritis, atherosclerosis, coronary heart disease, and tumor formation. Therefore, the apricot is one of the many beneficial stone fruits, regardless of being consumed fresh or dried [1].

Although the use of chemical fertilizers can lead to increased crop productivity, it has undesirable effects on the environment [2]. Moreover, the long-term and excessive application of chemical fertilizers can considerably reduce soil pH, which leads to decreased bacterial diversity and negative impacts on the physical and chemical properties of the soil, such as soil compaction, degradation, and acidification, resulting in decreased soil organic matter and fertility [3–5]. Additionally, the application of chemical fertilizers has passive effects on fruit quality attributes, seriously impacting the environment through, for example, nitrogen leaching and ground water pollution [6–8].

A plant biostimulant is defined as any substance or microorganism, which, when it is applied to plants, can increase the nutrition efficacy, nutrients' uptake, yield, fruit quality

characteristics, and tolerance of abiotic stresses [9–12]. The application of biostimulants is considered a good suitable to reduce the use of chemical fertilizers. Biostimulants are an essential source of macro- and microelements and plant hormones such as auxins, cytokinins, and gibberellins; they also are safe for the soil and do not harmfully affect soil characteristics. Moreover, they are cheap and easy to prepare and use [13].

Humic acid (HA) is a hydrophilic colloid, a polymer mixture with negative charge that can interact with many organic and inorganic substances due to its weak acidity, hydrophilicity, colloid, adsorption, ion exchange, complexation, redox, and physiological activity. Moreover, it is an organic biostimulant that can markedly increase plant growth, crop productivity [14], and nutrient absorption [15–17]. According to de Melo [18], HA is soluble in alkaline media or water and can help alleviate the effects of pollution. Additionally, HA can accelerate cell division and improve the absorption and uptake of nutrients and water [19]. HA can increase the ability of plants to tolerate abiotic stresses, which consequently leads to an increasing plant growth and final yield [16,20]. As HA can stimulate cell respiration, photosynthesis process, enzyme activation, water absorption, and improve the availability and transport of elements in plants, as well as the composition, cation exchange capacity, and water maintenance ability of soil, it can encourage plant growth and increase the plant yield [21,22].

The brassinosteroids (Brs) are a group of plant hormones with high growth stimulatory effect, and they exist at little quantities in pollen grains, anthers, seeds, leaves, stems, roots, flowers, grain, and young vegetative tissues of plants, and their chemical composition is $C_{28}H_{48}O_6$. Additionally, it is a group of plant steroid hormones that have paramount roles in regulating various processes, such as plant growth, cell division, multiplication, flower formation, fruit ripening, and tolerance to biotic and abiotic stresses [23–27]. The Brs play a vital role in the sustainable production of horticultural crops as alternatives to lessen the excessive utilization of chemical fertilizers [28]. Additionally, the Brs are natural, nontoxic, biosafe, and ecofriendly phytohormones, so they are important for improving the growth, development, productivity, and produce quality of crop plants [29–31]. These steroidal compounds are also involved in improving the defense mechanism of plants against different biotic and abiotic stresses, such as water, temperature, oxidative, and high salinity stress. Moreover, they can regulate numerous physiological and developmental processes in plants [32,33] and can improve fruit productivity [31] by regulating plant growth, the flowering process, fruit set, cell division elongation, and enzyme activation through involvement in numerous processes such as nucleic acid and protein syntheses and photosynthesis [31,34].

Seaweed extract (SWE) is characterized by high amounts of plant growth regulators like auxins and cytokinins, vitamins, amino acids, organic matter, polysaccharides, and sterols, so it is considered a plant growth stimulant [35]. As SWE contains natural compounds that promote nutrient availability and their absorption, it can also increase plant growth and yield [36]. Additionally, SWE increases the seed germination rate, plant development, and biotic, and abiotic stress toleration and lengthens the shelf life of the produce [37]. Moreover, the foliar application of SWE positively influences the growth, productivity, and fruit quality attributes and nutritional status of fruit trees [38–40]. Since SWE is rich in minerals as P, K, Ca, Mg, Cu, Fe, Mn, and Zn, it has the ability to increase growth attributes, the yield, and its components, as well as the fruit content of the protein and leaf mineral content of nitrogen and phosphorous under semiarid and desert conditions [41].

Therefore, the aim of this study is to decrease the dependency on the usage of chemical fertilizers in the apricot trees' nutrition by using ecofriendly and sustainable biostimulants such as HA, Brs, and SWE.

## 2. Materials and Methods

### 2.1. Experimental Site and Treatments

The current study was conducted during the years 2021 and 2022 to investigate the effect of foliar sprays containing three different levels of three different biostimulants on

the vegetative growth parameters, yield, fruit quality characteristics, and mineral content in the leaves of apricot (*Prunus armeniaca)* cv. Canino. The experiment was performed in at the Bader Centre in the El Beheira Governorate, Egypt, which is located at latitude 30.57746, longitude 30.71573. The experiment included ten treatments, and each treatment contained six trees (six replicates). Sixty trees were selected that were similar in their age (8 years), vigor, growth, and size, and they were pruned uniformly to an open center shape. The trees were budded on Mariana rootstock and planted in sandy soil under a drip irrigation system 4 × 4 m apart. All the horticultural practices that were applied in the orchard were uniform, except foliar spraying. Each tree was fertilized with 10 kg organic manure + 1 kg ammonium nitrate + 1 kg monocalcium phosphate + 0.5 kg potassium sulfate. The results of the physical and chemical analyses of the soil, performed according to Parikh [42], are shown in Table 1. The chilling hours are shown in Table 2, and the climate data are shown in Table 3.

**Table 1.** Physicochemical characteristics of the experimental soil.

| Clay (%) | Silt (%) | Sand (%) | Soil Texture | Sum of Bases |
|---|---|---|---|---|
| **12.2** | 22.80 | 65 | Sandy loam | 9.33 meq/L |
| **pH (1 Soil:1 H$_2$O)** | **EC dsm$^{1-}$ (1 Soil:1 H$_2$O)** | **Total CaCO$_3$$^{2-}$** | **Organic Matter** | **Cation exchange capacity (CEC)** |
| **7.9** | 0.94 | 2.23% | 1.46% | 13.39 meq/100 g soil |
| **Available Macronutrients (g/kg soil)** | | | | |
| **N** | **P** | | **K** | |
| **0.149** | 0.019 | | 0.698 | |
| **Soluble Anions (%)** | | | | |
| **HCO$_3$$^-$** | **Cl$^-$** | | **SO$_4$$^{2-}$** | |
| **3.96** | 2.45 | | 2.65 | |
| **Soluble Cations (%)** | | | | |
| **Na$^+$** | **Mg$^{2+}$** | **K$^+$** | | **Ca$^{2+}$** |
| **2.05** | 1.80 | 2.38 | | 3.10 |

**Table 2.** Chilling hours from December 2020 to March 2022.

| Months December 2020 to March 2022 | Chilling Hours during 2020–2021 | | | Chilling Hours during 2021–2022 | | |
|---|---|---|---|---|---|---|
| | **Total Hours in Season** | **Chilling Total Hours** | **Chilling %** | **Total Hours in Season** | **Chilling Total Hours** | **Chilling %** |
| | 2904 | 390 | 13.43 | 2904 | 965 | 33.23 |

**Table 3.** Climate classification and climate data during the 2021–2022 seasons.

| Months | 2021 | | | | 2022 | | | |
|---|---|---|---|---|---|---|---|---|
| | **Average Temperature (°C)** | **Average Relative Humidity (%)** | **Precipitation (mm)** | **Average Wind Speed (m/s)** | **Average Temperature (°C)** | **Average Relative Humidity (%)** | **Precipitation (mm)** | **Average Wind Speed (m/s)** |
| **January** | 14.61 | 63.97 | 4.50 | 2.57 | 11.11 | 68.41 | 35.10 | 2.60 |
| **February** | 14.60 | 64.92 | 28.80 | 2.24 | 12.74 | 67.33 | 9.10 | 2.48 |
| **March** | 15.88 | 63.22 | 81.40 | 2.75 | 13.75 | 62.39 | 25.40 | 2.89 |
| **April** | 19.92 | 53.97 | 0.40 | 2.98 | 22.19 | 46.60 | 0.80 | 3.16 |
| **May** | 26.63 | 44.08 | 0.00 | 2.90 | 24.87 | 46.54 | 3.80 | 3.36 |
| **June** | 27.74 | 47.40 | 0.00 | 3.24 | 28.85 | 48.54 | 1.40 | 3.33 |
| **July** | 30.53 | 48.29 | 0.20 | 3.01 | 29.73 | 49.53 | 2.00 | 3.20 |
| **Average** | 21.42 | 55.12 | 115.30 | 2.81 | 20.46 | 55.62 | 77.60 | 3.00 |

During both years (2021 and 2022), the following treatments were applied to the trees: control (untreated trees), humic acid (HA) (Shandong Jinrunzi Bio-Tech Co., Ltd., Binzhou, China) at 500, 1000, or 2000 mg/L; brassinosteroids (Brs) (Shanghai CIE Chemical Co., Ltd., Shanghai, China) at 0.5, 1, or 2 mg/L; and seaweed extract (SWE) (Qingdao Haidelong Biotechnology Co., Ltd., Qingdao, China) at 1000, 2000, or 3000 mg/L. The composition of SWE is 16% alginic acid, 50 % organic matter, 1% N, 16–21 $K_2O$, 600–800 mg/L cytokinin and gibberellin, 1–6% mannitol, 0.2% Fe, 0.15% Ca, 0.2% Mg, and 1% S. The treatments were applied four times starting from the swelling bud stage (2nd week of February), at the balloon stage (in March), just after fruit set, and one month before harvest. At the start of February, eight branches from each tree (replicate) were selected that were distributed well around the trees and labeled. Each treatment included six trees.

### 2.2. Vegetative Parameters

After picking the fruits in July both years, the shoot length was measured in centimeters. The leaf total chlorophyll content was measured with a chlorophyll meter (SPAD-502, Minolta Co., Tokyo, Japan) Yadava [43] by taking the average of 15 readings from each tree (replicate) in each treatment. The leaf area ($cm^2$) was determined according to Equation (1) [44].

$$\text{Leaf area (cm}^2) = [0.49 \text{ (length of leaf} \times \text{width of the leaf)} + 19.69] \tag{1}$$

### 2.3. Fruit Set and Yield

The fruit set (%) was estimated according to formula (2).

$$\text{Fruit set \%} = \frac{\text{Number of fruitlets}}{\text{Total number of flowers}} \times 100 \tag{2}$$

The yield (kg/tree) was estimated for each replicate/tree in June each year, whereas the fruit in tons per unit area (hectare) was estimated by multiplying the average of the tree yield with the number of trees in one hectare.

### 2.4. Fruit Quality

2.4.1. Fruit Physical Characteristics

Twenty fruits from each tree (replicate) in each treatment were randomly chosen, and then, the averages of their fruit weight (g), fruit length (cm), and diameter (cm) were measured with hand vernier calipers. The fruit size was estimated by weighing the removed water. The fruit firmness (Ib/inch$^2$) was estimated using a Magness and Taylor [45] pressure tester with a 5/16 inch plunger (mod. FT 02) (0–2 Lb., Via Reale, 63, 48,011 Alfonsine, Italy).

2.4.2. Fruit Chemical Characteristics

The total soluble solids (TSS %) in the apricot fruits were determined with a hand refractometer. Fruit acidity was calorimetrically measured based on the estimated malic acid using five milliliters of fruit juice and titrated with 0.1 N NaOH of a known normality using phenolphthalein as an indicator [46]. Vitamin C (mg/100 mL juice) was measured using 3% oxalic acid and 2,6-dichlorophenol indophenols [47]. The total sugars and reducing sugar percentages were calorimetrically determined using phenol and sulfuric acid extracted from 5 g of fresh pulp [48]. Nonreducing sugar contents were calculated by the difference between the total and reducing sugars.

### 2.5. Mineral Content in Apricot Leaves

Twenty leaves from each tree were collected after picking the fruits in July 2021 and 2022, as mentioned by Arrobas et al. [49], to determine their mineral contents of the macro- and micronutrients. The leaves were washed with tap water and then with distilled water. Next, they were dried in an oven at 70 °C until a consistent weight and crushed. The samples were digested with $H_2SO_4$ and $H_2O_2$. The nitrogen content was measured

following the micro-Kjeldahl method, as described by Wang et al. [50], and phosphorus by the vanadomolybdate method, as cited by Wieczorek et al. [51], with a spectrophotometer at a wavelength of 405 nm. The potassium was measured with a flame photometer (SKZ International Co., Ltd., Jinan Shandong, China) [52]. By using atomic absorption (3300), leaf Ca at 422.8 nm, Mg at 285.2 nm, Fe at 248.3 nm, Zn at 213.9 nm, and Mn at 279.5 nm were determined as described by Stafilov and Karadjova [53].

### 2.6. Statistical Analysis

All the obtained data in 2021 and 2022 were subjected to an analysis of variance (ANOVA) for a randomized complete block design (RCBD) by using Duncan's test at 0.05, followed by a comparison of the means with the least significant difference at a probability 5% according to Snedecor and Cochran [54], which was measured with CoHort Software (Pacific Grove, CA, USA).

## 3. Results

### 3.1. Vegetative Growth Parameters

Table 4 demonstrates that spraying of SWE, Brs, and HA considerably improved the vegetative growth attributes shoot length, leaf area, and leaf total chlorophyll content compared with untreated trees during both years of the study. The highest increases were noticed with the foliar application of 3000 mg/L SWE, 2 mg/L Brs, and 2000 mg/L HA compared with the application of 1000 mg/L SWE, 0.5 mg/L Brs, and 500 mg/L HA in both experimental years. Moreover, the leaf chlorophyll content significantly increased with the application of 3000 mg/L SWE compared with spraying 2000 or 1000 mg/L in both 2021 and 2022. We also found that the application of 2 mg/L Brs or 2000 mg/L HA significantly increased the leaf total chlorophyll compared with the application of 1 or 0.5 mg/L Brs or 500 mg/L HA, respectively, for both experimental years.

**Table 4.** Effect of foliar spraying of HA, Brs, and SWE on shoot length, leaf area, and leaf chlorophyll content for apricot cv. Canino during 2021–2022.

| Treatments | | Shoot Length (cm) | | Leaf Area (cm$^2$) | | Chlorophyll Content (SPAD) | |
|---|---|---|---|---|---|---|---|
| | | 2021 | 2022 | 2021 | 2022 | 2021 | 2022 |
| **Control** | 0 | 37.53e ±0.44 | 38.16e ±0.60 | 31.15e ±0.55 | 31.62 ±0.62 | 43.60g ±0.41 | 44.37e ±0.56 |
| **HA** | 500 mg/L | 39.49d ±0.60 | 40.94d ±0.96 | 32.95cde ±0.25 | 33.57cd ±0.72 | 47.06ef ±0.61 | 47.93d ±0.43 |
| | 1000 mg/L | 42.45bc ±0.87 | 43.40bc ±0.45 | 34.47bcd ±0.56 | 35.86bc ±0.18 | 48.60de ±0.90 | 50.08c ±0.89 |
| | 2000 mg/L | 43.42ab ±0.33 | 45.11ab ±0.65 | 37.03ab ±0.86 | 38.03ab ±0.47 | 51.64ab ±0.57 | 52.50a ±0.50 |
| **Brs** | 0.5 mg/L | 40.28d ±0.87 | 41.70cd ±0.59 | 32.45de ±1.05 | 33.73cd ±0.85 | 46.89ef ±0.22 | 46.37d ±0.59 |
| | 1 mg/L | 41.25cd ±0.59 | 43.47bc ±0.31 | 35.55abc ±0.69 | 36.58ab ±0.38 | 48.97cde ±0.81 | 47.95d ±0.42 |
| | 2 mg/L | 43.24ab ±0.39 | 44.73ab ±0.31 | 36.55ab ±1.19 | 38.21ab ±1.15 | 50.90abc ±1.16 | 52.07ab ±1.02 |
| **SWE** | 1000 mg/L | 39.6d ±0.59 | 41.38d ±0.65 | 32.89cde ±0.87 | 34.24c ±0.74 | 46.26f ±0.58 | 46.89d ±0.85 |
| | 2000 mg/L | 42.26bc ±0.89 | 44.13b ±0.97 | 35.27abc ±0.92 | 36.90ab ±0.57 | 49.83bcd ±0.90 | 50.78bc ±0.10 |
| | 3000 mg/L | 44.88a ±0.33 | 46.32a ±1.00 | 37.73a ±0.58 | 38.72a ±0.82 | 52.78a ±0.87 | 53.57a ±0.72 |
| **LSD $_{0.05}$** | | 1.70 | 182 | 2.48 | 2.17 | 2.07 | 1.60 |

In each column, treatments with the same letters had insignificant differences among them by using Duncan's test at 0.05.

*3.2. Fruit Set Percentage, and Yield*

Table 5 illustrates that the foliar application of HA, Brs, and SWE positively improved the fruit set percentage and fruit yield per tree and per unit as compared with those of untreated trees during 2021–2022. The fruit set percentage and fruit yield markedly increased with the spraying of 3000 mg/L SWE, 2 mg/L Brs, and 2000 mg/L HA during the two years compared with the application of 1000 or 2000 mg/L SWE, 0.5 or 1 mg/L Brs, and 500 or 1000 mg/L HA during both years. Moreover, the effect of HA, Brs, and SWE strengthened with increasing the application concentration.

**Table 5.** Effect of foliar spraying of HA, Brs, and SWE on the fruit set percent, fruit yields in kilograms, or tons of apricot cv. Canino during 2021–2022.

| Treatment | | Fruit Set % | | Fruit Yield/Tree (kg) | | Fruit Yield/Hectare (tons) | |
|---|---|---|---|---|---|---|---|
| | | 2021 | 2022 | 2021 | 2022 | 2021 | 2022 |
| **Control** | 0 | 20.43f ±1.01 | 21.53e ±0.88 | 36.52f ±0.74 | 38.07f ±0.70 | 21.91f ±0.44 | 22.84f ±0.42 |
| **HA** | 500 mg/L | 22.43ef ±0.50 | 24.15de ±1.15 | 38.96de ±0.83 | 39.80ef ±0.91 | 23.37de ±0.50 | 23.88ef ±0.55 |
| | 1000 mg/L | 25.98cd ±1.00 | 25.13cd ±0.66 | 40.96cd ±0.49 | 42.55d ±0.38 | 24.57cd ±0.29 | 25.53d ±0.23 |
| | 2000 mg/L | 28.95ab ±0.87 | 30.50ab ±1.42 | 43.24ab ±0.67 | 45.14abc ±0.65 | 25.95ab ±0.40 | 27.08abc ±0.39 |
| **Brs** | 0.5 mg/L | 23.45de ±0.61 | 23.05de ±0.65 | 38.50ef ±1.19 | 38.36ef ±1.17 | 23.10ef ±0.72 | 23.01ef ±0.70 |
| | 1 mg/L | 25.04cde ±0.66 | 25.75cd ±0.56 | 39.78cde ±0.63 | 42.97cd ±0.47 | 23.87cde ±0.38 | 25.78cd ±0.28 |
| | 2 mg/L | 29.30a ±0.79 | 30.20ab ±1.02 | 43.23ab ±0.81 | 45.70ab ±0.98 | 25.94ab ±0.48 | 27.42ab ±0.59 |
| **SWE** | 1000 mg/L | 23.78cde ±0.74 | 24.64d ±1.18 | 38.76ef ±0.98 | 40.45e ±1.32 | 23.26e ±0.59 | 24.27e ±0.79 |
| | 2000 mg/L | 26.45bc ±0.59 | 27.69bc ±0.57 | 41.46bc ±1.02 | 43.52bcd ±0.45 | 24.88bc ±0.61 | 26.11bcd ±0.27 |
| | 3000 mg/L | 31.48a ±1.18 | 32.97a ±1.17 | 43.97a ±0.93 | 46.92a ±0.76 | 26.38a ±0.56 | 28.15a ±0.46 |
| **LSD $_{0.05}$** | | 2.55 | 2.73 | 1.99 | 2.09 | 1.19 | 1.25 |

In each column, treatments with same letters were insignificantly different by using Duncan's test at 0.05.

*3.3. Fruit Quality*

3.3.1. Physical Fruit Characteristics

Fruit physical characteristics in terms of the fruit weight, fruit size, fruit length, fruit diameter, and fruit firmness were remarkably increased with the application of HA, Brs, and SWE compared with those of unsprayed trees (Table 6). We noticed that spraying with 3000 mg/L SWE, 2 mg/L Brs, and 2000 mg/L HA produced significantly different results from the application of 1000 mg/L SWE, 0.5 mg/L Brs, and 500 mg/L HA for the same measured fruit quality attributes during both experimental years. Moreover, the application of 3000 mg/L SWE, 2 mg/L Brs, and 2000 mg/L HA increased the measured treatments, but for some parameters, they produced insignificant differences compared with the results of the application of 2000 mg/L SWE, 1 mg/L Brs, or 1000 mg/L HA during the two years.

**Table 6.** Effect of foliar spraying of HA, Brs, and SWE on the fruit weight, size, length, diameter, and firmness of apricot cv. Canino during 2021–2022.

| Treatments | | Fruit Weight (g) | | Fruit Size (cm³) | | Fruit Length (cm) | | Fruit Diameter (cm) | | Fruit Firmness (Ib/inch²) | |
|---|---|---|---|---|---|---|---|---|---|---|---|
| | | **2021** | **2022** | **2021** | **2022** | **2021** | **2022** | **2021** | **2022** | **2021** | **2022** |
| **Control** | 0 | 26.97d ±071 | 26.48f ±0.50 | 38.40e ±0.58 | 40.01d ±0.25 | 2.88f ±0.08 | 2.91c ±0.03 | 3.01g ±0.05 | 3.04d ±0.05 | 11.00e ±0.29 | 10.87e ±0.24 |
| **HA** | 500 mg/L | 27.24d ±0.96 | 29.90e ±0.69 | 41.30d ±0.61 | 44.17c ±0.52 | 3.06e ±0.06 | 3.08c ±0.06 | 3.15fg ±0.03 | 3.07cd ±0.03 | 12.03cd ±0.28 | 12.40d ±0.11 |
| | 1000 mg/L | 32.19b ±1.29 | 33.49cd ±0.60 | 46.63bc ±1.22 | 47.97b ±0.55 | 3.25d ±0.03 | 3.69b ±0.09 | 3.31ef ±0.05 | 3.25c ±0.06 | 12.11cd ±0.05 | 13.00c ±0.23 |
| | 2000 mg/L | 34.61ab ±0.86 | 35.40abc ±0.65 | 48.97ab ±0.60 | 48.87b ±0.64 | 3.74b ±0.04 | 3.85ab ±0.09 | 3.74b ±0.07 | 3.93a ±0.01 | 12.17bcd ±0.1 | 13.61ab ±0.01 |
| **Brs** | 0.5 mg/L | 27.23d ±1.20 | 30.02e ±1.29 | 40.30de ±1.23 | 43.83c ±2.32 | 2.89f ±0.03 | 3.09c ±0.06 | 3.05g ±0.06 | 3.13cd ±0.05 | 11.70de ±0.15 | 12.33d ±0.18 |
| | 1 mg/L | 31.85bc ±0.82 | 34.84bc ±0.60 | 45.57c ±0.64 | 48.46b ±0.64 | 3.40c ±0.03 | 3.77b ±0.06 | 3.35de ±0.08 | 3.58b ±0.01 | 12.16bcd ±0.04 | 12.97c ±0.27 |
| | 2 mg/L | 34.97ab ±0.84 | 37.62ab ±0.90 | 48.00abc ±1.03 | 51.23ab ±0.99 | 3.91a ±0.05 | 3.85ab ±0.10 | 3.57bc ±0.06 | 3.91a ±0.01 | 12.93ab ±0.30 | 13.77a ±0.14 |
| **SWE** | 1000 mg/L | 29.00cd ±0.94 | 31.32de ±0.93 | 42.00d ±0.94 | 44.37c ±0.55 | 3.02e ±0.06 | 3.04c ±0.02 | 3.09g ±0.06 | 3.17d ±0.03 | 11.80d ±0.11 | 12.37d ±0.18 |
| | 2000 mg/L | 33.53ab ±0.35 | 35.69abc ±0.60 | 46.60bc ±0.30 | 49.20b ±0.61 | 3.42c ±0.01 | 3.76b ±0.03 | 3.53cd ±0.14 | 3.61b ±0.09 | 12.13cd ±0.18 | 13.13bc ±0.13 |
| | 3000 mg/L | 35.4a ±0.87 | 38.14a ±1.48 | 49.80a ±0.53 | 52.98a ±1.55 | 3.97a ±0.06 | 4.04a ±0.06 | 4.00a ±0.06 | 4.00a ±0.08 | 13.07a ±0.52 | 13.87a ±0.12 |
| **LSD 0.05** | | 2.86 | 2.68 | 2.51 | 3.07 | 0.13 | 0.21 | 0.18 | 0.15 | 0.74 | 0.54 |

In each column, treatments with the same letters were not significantly different by using Duncan's test at 0.05.

### 3.3.2. Fruit Chemical Characteristics

Table 7 shows that foliar spraying with 3000 mg/L SWE, 2 mg/L Brs, and 2000 mg/L HA considerably improved the fruit content of total soluble solids and vitamin C compared with the application of 1000 or 2000 mg/L SWE, 0.5 or 1 mg/L Brs, and 500 or 1000 mg/L HA during both experimental years. The results also indicated that the effect of SWE, Brs, and HA gradually increased with increasing the applied concentration. However, for fruit acidity, the opposite trend was observed. The same applied treatments significantly reduced the fruit acidity during 2021 and 2022.

**Table 7.** Effect of foliar spraying of HA, Brs, and SWE on the fruit content from the total soluble solids, vitamin C, and total acidity of apricot cv. Canino during 2021–2022.

| Treatments | | TSS (%) | | Vitamin C (mL/100 mL) | | Total Acidity (%) | |
|---|---|---|---|---|---|---|---|
| | | **2021** | **2022** | **2021** | **2022** | **2021** | **2022** |
| **Control** | 0 | 11.16e ±0.3 | 10.83d ±0.32 | 12.50f ±0.25 | 12.59d ±0.16 | 0.76a ±0.01 | 0.75a ±0.01 |
| **HA** | 500 mg/L | 12.42d ±0.23 | 11.43d ±0.22 | 13.73de ±0.54 | 14.33bc ±0.22 | 0.68bcd ±0.01 | 0.72a ±0.01 |
| | 1000 mg/L | 13.17bc ±0.14 | 12.13c ±0.24 | 15.13bc ±0.32 | 15.21b ±0.47 | 0.65de ±0.01 | 063b ±0.01 |
| | 2000 mg/L | 14.76a ±0.17 | 13.97a ±0.27 | 17.10a ±0.20 | 16.87a ±0.57 | 0.60f ±0.01 | 0.60bc ±0.02 |
| **Brs** | 0.5 mg/L | 11.59e ±0.30 | 11.17d ±0.30 | 13.42ef ±0.28 | 14.56bc ±0.46 | 0.70b ±0.01 | 0.70a ±0.01 |
| | 1 mg/L | 12.72cd ±0.09 | 12.10c ±0.1 | 14.78bcd ±0.16 | 15.25b ±0.22 | 0.66bcde ±0.01 | 0.63b ±0.01 |
| | 2 mg/L | 14.66a ±0.11 | 13.14b ±0.18 | 16.51a ±0.38 | 16.99a ±0.50 | 0.63ef ±0.02 | 0.57c ±0.01 |
| **SWE** | 1000 mg/L | 12.32d ±0.24 | 11.37d ±0.12 | 14.11cde ±0.26 | 13.56cd ±0.34 | 0.70bc ±0.01 | 0.70a ±0.01 |
| | 2000 mg/L | 13.59b ±0.35 | 12.57bc ±0.09 | 15.33b ±0.31 | 15.04b ±0.32 | 0.66cde ±0.01 | 0.61bc ±0.01 |
| | 3000 mg/L | 14.80a ±0.10 | 14.00a ±0.17 | 17.46a ±0.64 | 17.24a ±0.61 | 0.56g ±0.02 | 0.57c ±0.01 |
| **LSD 0.05** | | 0.59 | 0.64 | 1.06 | 1.25 | 0.04 | 0.04 |

In each column, treatments with the same letters were not significantly different by using Duncan's test at 0.05.

Table 8 reveals that treating apricot trees with 3000 mg/L SWE, 2 mg/L Brs, and 2000 mg/L HA significantly increased the fruit content of the total, reduced, and nonreduced sugars compared with those of untreated trees during 2021 and 2022. The effect of 3000 mg/L SWE was stronger than the effect of 2 mg/L Brs or 2000 mg/L HA in improving the fruit total, reduced, and nonreduced sugar contents during 2021 and 2022. The influence of HA was stronger at 2000 mg/L than at 1000 or 500 mg/L; the influence of Brs at 2 mg/L was stronger than at 1 or 0.5 mg/L during both years.

**Table 8.** Effect of foliar spraying of HA, Brs, and SWE on the fruit content of the total, reduced, and nonreduced sugars of apricot cv. Canino during 2021–2022.

| Treatments | | Total Sugars % | | Reduced Sugars % | | Nonreduced Sugars % | |
|---|---|---|---|---|---|---|---|
| | | **2021** | **2022** | **2021** | **2022** | **2021** | **2022** |
| **Control** | 0 | 7.66f | 7.63f | 5.11f | 5.09f | 2.55f | 2.55f |
| | | ±0.29 | ±0.27 | ±0.19 | ±0.18 | ±0.10 | ±0.09 |
| **HA** | 500 mg/L | 8.97de | 9.12d | 5.98de | 6.08d | 2.99de | 3.04d |
| | | ±0.17 | ±0.25 | ±0.11 | ±0.16 | ±0.06 | ±0.08 |
| | 1000 mg/L | 9.48cd | 10.37c | 6.32cd | 6.91c | 3.16cd | 3.46c |
| | | ±0.29 | ±0.20 | ±0.19 | ±0.13 | ±0.09 | ±0.07 |
| | 2000 mg/L | 11.32a | 11.13b | 7.54a | 7.42b | 3.77a | 3.71b |
| | | ±0.14 | ±0.26 | ±0.09 | ±0.17 | ±0.05 | ±0.09 |
| **Brs** | 0.5 mg/L | 8.50e | 8.06e | 5.66e | 5.37e | 2.84e | 2.69e |
| | | ±0.36 | ±0.27 | ±0.24 | ±0.18 | ±0.12 | ±0.09 |
| | 1 mg/L | 9.83bcd | 10.46c | 6.55bcd | 6.97c | 3.28bcd | 3.48c |
| | | ±0.26 | ±0.07 | ±0.17 | ±0.05 | ±0.09 | ±0.02 |
| | 2 mg/L | 10.38b | 11.32b | 6.92b | 7.55b | 3.46b | 3.77b |
| | | ±0.31 | ±0.32 | ±0.20 | ±0.22 | ±0.10 | ±0.10 |
| **SWE** | 1000 mg/L | 9.40cd | 8.35e | 6.27cd | 5.57e | 3.13cd | 2.78e |
| | | ±0.15 | ±0.29 | ±0.10 | ±0.20 | ±0.05 | ±0.10 |
| | 2000 mg/L | 10.14bc | 10.59c | 6.76bc | 7.06c | 3.38bc | 3.53c |
| | | ±0.26 | ±0.30 | ±0.17 | ±0.20 | ±0.09 | ±0.10 |
| | 3000 mg/L | 11.44a | 11.83a | 7.62a | 7.89a | 3.82a | 3.94a |
| | | ±0.29 | ±0.31 | ±0.19 | ±0.21 | ±0.10 | ±0.10 |
| **LSD $_{0.05}$** | | 0.80 | 0.41 | 0.53 | 0.28 | 0.27 | 0.14 |

In each column, treatments with the same letters were not significantly different by using Duncan's test at 0.05.

### 3.4. Leaf Mineral Content of Macro- and Micronutrients

The results in Table 9 show that the application of 3000 mg/L SWE, 2 mg/L Brs, and 2000 mg/L HA markedly improved the leaf contents of N, P, K, Ca, and Mg compared with those with individual spraying of 1000 mg/L SWE, 0.5 mg/L Brs, or 500 mg/L HA and untreated trees during the two experimental years. Furthermore, the effect of foliar spraying with SWE, Brs, and HA strengthened with increasing the applied concentration by each during the two years. The influence of 3000 mg/L SWE was significantly different from the application of 2000 or 1000 mg/L SWE on improving the leaf mineral content of macronutrients such as N, P, K, Ca, and Mg during both experimental years. The highest increases were found for the application of SWE at 3000 mg/L compared with 2000 mg/L HA or 2 mg/L Brs, but the differences were insignificant during the two years.

**Table 9.** Effect of foliar spraying of HA, Brs, and SWE on the leaf mineral content of macronutrients of apricot cv. Canino during 2021–2022.

| Treatments | | N % | | P % | | K % | | Ca % | | Mg % | |
|---|---|---|---|---|---|---|---|---|---|---|---|
| | | 2021 | 2022 | 2021 | 2022 | 2021 | 2022 | 2021 | 2022 | 2021 | 2022 |
| **Control** | 0 | 2.28d ±0.17 | 2.40g ±0.06 | 0.33e ±0.01 | 0.34g ±0.01 | 2.19e ±0.01 | 2.26f ±0.04 | 1.62d ±0.05 | 1.73f ±0.04 | 0.60e ±0.03 | 0.64e ±0.01 |
| **HA** | 500 mg/L | 2.64c ±0.09 | 2.67ef ±0.02 | 0.37d ±0.01 | 0.37f ±0.01 | 2.33de ±0.14 | 2.37e ±0.04 | 1.77c ±0.03 | 1.81e ±0.02 | 0.69cd ±0.02 | 0.69cd ±0.02 |
| | 1000 mg/L | 2.68c ±0.04 | 2.79de ±0.05 | 0.41bc ±0.01 | 0.40de ±0.01 | 2.41bcd ±0.07 | 2.56cd ±0.02 | 1.88b ±0.02 | 1.91cd ±0.02 | 0.72abc ±0.01 | 0.72bcd ±0.01 |
| | 2000 mg/L | 2.83abc ±0.09 | 3.10ab ±0.06 | 0.44ab ±0.01 | 0.44ab ±0.01 | 2.58b ±0.06 | 2.67b ±0.05 | 2.02a ±0.02 | 2.03b ±0.02 | 0.75ab ±0.01 | 0.80a ±0.01 |
| **Brs** | 0.5 mg/L | 2.63c ±0.06 | 2.60f ±0.06 | 0.38cd ±0.01 | 0.38ef ±0.01 | 2.33de ±0.01 | 2.46de ±0.02 | 1.74c ±0.02 | 1.87de ±0.01 | 0.66cd ±0.01 | 0.68cde ±0.01 |
| | 1 mg/L | 2.80bc ±0.06 | 2.99bc ±0.07 | 0.40cd ±0.01 | 0.41cd ±0.01 | 2.58b ±0.01 | 2.63bc ±0.05 | 1.87b ±0.03 | 1.97bc ±0.01 | 0.68cd ±0.02 | 0.73bc ±0.02 |
| | 2 mg/L | 3.00ab ±0.06 | 3.10ab ±0.06 | 0.41bc ±0.01 | 0.43abc ±0.01 | 2.77a ±0.01 | 2.68b ±0.03 | 2.01a ±0.02 | 2.03b ±0.02 | 0.74ab ±0.02 | 0.77ab ±0.01 |
| **SWE** | 1000 mg/L | 2.65c ±0.03 | 2.72def ±0.06 | 0.38cd ±0.01 | 0.37f ±0.01 | 2.38cd ±0.04 | 2.42e ±0.02 | 1.79c ±0.02 | 1.87de ±0.01 | 0.65d ±0.01 | 0.67de ±0.02 |
| | 2000 mg/L | 2.80bc ±0.03 | 2.88cd ±0.06 | 0.41bc ±0.01 | 0.43bc ±0.01 | 2.54bc ±0.11 | 2.63bc ±0.05 | 1.88b ±0.01 | 1.99b ±0.01 | 0.70bcd ±0.1 | 0.75ab ±0.02 |
| | 3000 mg/L | 3.06a ±0.03 | 3.20a ±0.02 | 0.45a ±0.01 | 0.45a ±0.01 | 2.81a ±0.04 | 2.80a ±0.02 | 2.08a ±0.01 | 2.11a ±0.02 | 0.77a ±0.01 | 0.80a ±0.02 |
| **LSD 0.05** | | 0.24 | 0.16 | 0.03 | 0.02 | 0.16 | 0.10 | 0.08 | 0.07 | 0.05 | 0.05 |

In each column, treatments with the same letters were not significantly different by using Duncan's test at 0.05.

The results in Table 10 indicate that the spraying of HA, Brs, and SWE increased the leaf mineral content of micronutrients such as Fe, Zn, and Mn compared with those of the control (untreated trees) during the years 2021, and 2022. The results also showed that the application of 3000 mg/L SWE, 2 mg/L Brs, and 2000 mg/L HA produced the largest increases in the leaf mineral contents of these micronutrients compared with the application of 1000 mg/L SWE, 0.5 mg/L Brs, and 500 mg/L HA during the study years. Furthermore, the effect of the application of 3000 mg/L SWE significantly differed from that of 2000 or 1000 mg/L. Moreover, the influence of 2 mg/L Brs and 2000 mg/L HA was much stronger than that of 1 or 0.5 mg/L Brs and 500 or 1000 mg/L HA during both experimental years.

**Table 10.** Effect of foliar spraying of HA, Brs, and SWE on the leaf mineral content of micronutrients of apricot cv. Canino during 2021–2022.

| Treatments | | Fe ppm | | Zn ppm | | Mn ppm | |
|---|---|---|---|---|---|---|---|
| | | 2021 | 2022 | 2021 | 2022 | 2021 | 2022 |
| **Control** | 0 | 100.4d ±1.25 | 102.07e ±0.93 | 20.21e ±1.16 | 22.43d ±0.91 | 31.52d ±0.64 | 31.63e ±1.02 |
| **HA** | 500 mg/L | 104.33c ± 1.03 | 105.63cde ±1.29 | 23.28cd ±1.46 | 24.12cd ±0.79 | 34.54c ±0.84 | 34.06de ±0.89 |
| | 1000 mg/L | 108.37b ±0.32 | 108.67bcd ±2.24 | 27.51c ±0.60 | 26.67c ±0.63 | 37.74b ±0.44 | 39.25bc ±0.89 |
| | 2000 mg/L | 110.7b ±1.10 | 111.83ab ±0.78 | 30.30a ±0.26 | 31.09b ±1.43 | 40.48a ±0.97 | 41.79ab ±0.54 |
| **Brs** | 0.5 mg/L | 103.07c ±0.12 | 104.67de ±1.41 | 22.10de ±0.98 | 23.27d ±0.99 | 34.82c ±0.49 | 34.48d ±0.88 |
| | 1 mg/L | 105.43c ±0.47 | 106.57cde ±2.14 | 25.39bc ±0.32 | 26.48c ±0.91 | 38.92ab ±1.14 | 38.27c ±0.32 |
| | 2 mg/L | 108.60b ±0.98 | 113.47ab ±1.91 | 30.80a ±1.16 | 29.38b ±1.39 | 39.72ab ±0.82 | 42.04ab ±0.92 |
| **SWE** | 1000 mg/L | 104.4c ±0.93 | 106.03cde ±0.77 | 22.01de ±0.95 | 23.65d ±0.87 | 34.64c ±0.61 | 34.84d ±0.88 |
| | 2000 mg/L | 110.6b ±1.36 | 110.63bc ±1.47 | 26.94b ±0.86 | 26.70c ±0.40 | 37.37b ±0.89 | 39.19bc ±1.20 |
| | 3000 mg/L | 114.07a ±1.07 | 116.40a ±104 | 32.32a ±1.05 | 34.14a ±1.11 | 41.52a ±1.19 | 43.31a ±0.25 |
| **LSD 0.05** | | 2.58 | 4.55 | 2.40 | 2.45 | 2.50 | 2.66 |

In each column, treatments with the same letters were not significantly different by using Duncan's test at 0.05.

## 4. Discussion

The obtained results showed that HA, Brs, and SWE positively increased the shoot length, shoot diameter, leaf total chlorophyll, fruit set percentage, fruit yield, fruit physical and chemical characteristics, and leaf mineral composition from macro- and micronutrients comparing with the control during our study seasons.

Applying HA to the plants induces the development of the roots, leading to improvements in the soil composition and, thus, to increases in nutrient absorption [55–57]. Additionally, HA can substantially increase the nutrient content of plants and the fruit's contents of total sugars, amino acids, proteins, and phenolic compounds [58,59]. As HA can increase cell membrane stability, water uptake under osmotic stress, stress resistance, potassium absorption, proteins, and hormones synthesis, HA is considered a growth enhancer [15,60]. Additionally, HA directly works as a semi-hormonal compound [61] and has the ability to ameliorate the soil physical condition; enhances soil microbial metabolism; encourages the development of root and stem [62]; increases the uptake of nutrients by chelating them, and improves the membrane infiltration conservation characteristics [63]. Applying HA at 5 and 20 mg/L significantly increased the total chlorophyll content and TSS/acidity ratio, berry size, and, consequently, fruit yield but significantly lessened the fruit titratable acidity [64]. Spraying HA at 13% on grape cv. Askari at concentrations of 2.5, 5, and 7500 mg/L improved the fruit content from the TSS, TA, TSS–TA ratio, cluster weight, size, length and yield, and berry firmness compared to untreated plants. The highest values were noticed by the application of 2500 mg/L [65]. Similarly, spraying apple trees with HA at 0.5% significantly increased the growth parameters; fruit set percent; fruit yield, weight, size, and firmness; SSC percent; total and reducing sugar contents; and leaf mineral contents of the macronutrients. Additionally, it decreased the fruit drop and acidity percentages compared with those untreated trees when applied at 0.5% to apple [66] and at 0.2%, 0.3%, and 0.4% on pomegranates [67]. The application of HA at 0.15%, 0.30%, and 0.45% on 'Zebda' mango (*Mangifera indica*) trees increased the tree growth; flowering; yield, fruit quality; photosynthetic rate; nutrients; and phytohormones such as auxins, gibberellins, and cytokinins, but it reduced the abscisic acid content [68].

These results agree with those reported by Symons et al. [69], they found that Brs are necessary for plant growth and development, stem elongation, pollen tube development, fruit ripening, and productivity. Additionally, Brs were found to be involved in cell division, the formation of root systems, vegetative growth, the regulation of flowering time, fruit set, productivity, and fruit quality. Brs have the ability to increase the plant stress responses against abiotic stresses [70,71], such as temperature extremes [72]; water logging; drought [73]; salt stress [74]; and heavy metal toxicity such as cadmium, chromium, copper, nickel, and lead [75,76]) by inducing protein biosynthesis [77]. Applying Brs at 0.1 mg/L on yellow passion fruit increased the fruit number and fruit soluble solid content over those of untreated trees [78]. The use of Brs enhanced plant growth and lateral bud development; increased the number of flowers, fruit set, and number of fruits harvested; and decreased the flower and fruit drop rates. The foliar application of Brs encouraged fruit growth, increased the sweetness of fruits, and delayed the aging of leaves, all of which enhanced the quality of the produce [24]. Exogenous spraying of Brs affected the expression of genes related to the biosynthesis and transport of auxins and gibberellins [79,80] and increased the growth of apple trees in terms of plant height, internode length, leaf number and area, and shoot length and diameter [81]. Seadh et al. [82] noticed that applying of Brs induced flower bud development, growth, fruit setting fruit yield, and quality and minimized the drop percentages of flowers and fruits. Applying Brs to 'Tak Danehe Mashhad' sweet cherries at 0.25, 0.5, and 0.75 mg/L increased the fruit color by increasing fruit anthocyanin, ascorbic acid, and phenol content; fruit firmness and weight, diameter, and length; and fruit yield and quality and lengthened its shelf life [83]. Treating pear trees with Brs at 0.5 and 1.0 mg/L increased the fruit content of TSS and reduced the fruit acidity compared with those of untreated trees; 1 mg/L was the superior treatment [84]. Foliar application with 0.05 mg/L Brs increased plant growth and alleviated the negative effects of various stresses,

including water stress [85]. Spraying Brs at concentrations of 1, 10, and 50 mg/L resulted in statistically significant increases in flower and fruit numbers, elongation of pollen tubes, fruit set percentages, and nut weights compared with those of a control group [86].

The application of SWE promotes polyamines biosynthesis in fruits; increases cell division; and increases the contents of macro- and micronutrients, carbohydrates, and hormonal substances, especially cytokinins; this consequently increases the fruit weight and size [35]. Additionally, SWE can ameliorate the rate of photosynthesis and stomatal opening in plants [87]. In recent years, SWE has been applied on a wide scale to increase plant growth, productivity, and fruit quality attributes and reduce the chemical pollutants in the produce. SWEs are also characterized by high contents of cytokinins, auxins, gibberellins, vitamins, polysaccharides, micro- and macro elements, amino acids, and organic acids [88], so their application results in increased cell expansion and division, and, ultimately, shoot length and leaf area [89]. SWE is a rich source of phosphorus, potassium, calcium, magnesium, iron, manganese, zinc, selenium, iodine, cytokinins, IAA, $GA_3$, amino acids, vitamins, and antioxidants, so it can be considered a plant growth stimulator [90,91]. Sprayings of seaweed extract at 1.0 or 2.0 mL/L on strawberries increased the plant length, leaf number per plant, leaf area, root, carbohydrate, content, leaf phosphorus and potassium contents, fruit weight, and yield per plant. Additionally, the applied treatments also increased the fruit firmness, SSC, titratable acidity, and SSC–acidity ratio [92]. The application of seaweed extracts at 1% and 2% on 'Sukary' date palms increased the weights of bunches, fruits and fleshes, fruit yield, and soluble solids content and reduced the total sugars content, and the super concentration was 2% [93]. Treating the 'Gala' apple cultivar with various concentrations (0.1%, 0.2%, 0.3%, 0.4%, and 0.6%) of SWE via spraying resulted in significant improvements in the fruit set percent and fruit number, weight, and length compared with those of untreated trees. Additionally, the optimal concentration for achieving the maximum yield was 0.3%, which was higher than a previously reported concentration [94]. The foliar spraying of SWE at 0.3% or 0.4% increased the growth attributes, fruit set, fruit yield, physical and chemical properties of the fruit, and the tree nutritional content, compared with those of untreated trees [40].

### 5. Conclusions

From the obtained results, the application of HA, Brs, and SWE could be applied as safe and ecofriendly alternatives to reduce the reliance on the chemical fertilizers in apricot orchards to keep up the maintenance of fruit quality and soil fertility; they even have the ability to significantly improve the performance of apricot trees. This study supplies a foundation for the future clarification of HA, Brs, and SWE modified molecular mechanisms in apricot, which can make a remarkable addition to the scientific community.

**Author Contributions:** Conceptualization, W.F.A.M. and R.M.A.; methodology, W.F.A.M. and R.M.A.; software, W.F.A.M., A.M.A.-S. and L.S.-P.; validation, A.M.A.-S. and L.S.-P.; formal analysis, W.F.A.M., A.M.A.-S. and L.S.-P.; investigation, W.F.A.M. and R.M.A.; resources, W.F.A.M., A.M.A.-S. and R.M.A.; data curation, W.F.A.M., R.M.A. and L.S.-P.; writing—original draft preparation, W.F.A.M., R.M.A., A.M.A.-S. and L.S.-P.; writing—review and editing, W.F.A.M., R.M.A., A.M.A.-S. and L.S.-P.; and supervision, W.F.A.M., R.M.A., A.M.A.-S. and L.S.-P. All authors have read and agreed to the published version of the manuscript.

**Funding:** This research was funded by Researchers Supporting Project number (RSP2023R334), King Saud University, Riyadh, Saudi Arabia.

**Data Availability Statement:** All the required data are inserted in the manuscript.

**Acknowledgments:** The authors extend their appreciation to the Researchers Supporting Project number (RSP2023R334), King Saud University, Riyadh, Saudi Arabia.

**Conflicts of Interest:** The authors declare no conflict of interest.

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
