# Peer review of "Apricot (Prunus armeniaca) Performance under Foliar Application of Humic Acid, Brassinosteroids, and Seaweed Extract"

_horticulturae, doi:10.3390/horticulturae9040519_

Round 1

Reviewer 1 Report (Previous Reviewer 2)

Dear authors,

the manuscript requires reorganization in order to have a systematic flow and ultimately improve the clear understanding of the scientific message embedded in it. The discussion about the results is still very weak. The obtained results are not explained in it, but it only boils down to citing the literature. Too much literature is cited in the discussion.

Author Response

Reviewer 2 Report (Previous Reviewer 1)

The manuscript has been significantly improved, but needs some adjustments as shown in the attached file.

Author Response

Reviewer 3 Report (Previous Reviewer 3)

Dear Editor,

In this study, the authors have provided important information on the foliar application of Humic acid, Brassinoids and Seaweed extract, which would be help to enhance the audience knowledge and minimize the study gaps regarding bio-organic amendments such biostimulation. The manuscript is organized and informative. However, I am satisfied with author’s response and it could be accept.

Best wishes 

Author Response

Reviewer 4 Report (New Reviewer)

Comments to the manuscript (MS) "Apricot Performance under Foliar Application of Humic Acid, Brassinoids, and Seaweed Extract" (authors Adel M. Al-Saif , Lidia Sas-Paszt, Rehab M. Awad, Walid F. A. Mosa) submitted to the journal Horticulturae MDPI:

1. It is necessary to spent the time to revising and re-editing the text in order to ensure a higher quality of scientific analysis of the obtained data and their discussion.

2. ppm - would it be better to use traditional units? I would recommend it.

3. It is necessary to more clearly justify the categorization of the investigated products/substances using/discussing the relevant definitions of biostimulants, growth regulators, etc. 

For biostimulants, this is still an open and highly relevant topic, so each manuscript should provide detailed explanations/comments.

4. The authors of the manuscript work at Universities and Institutes in Saudi Arabia, Poland and Egypt. 

It is necessary to provide information on the regulation/categorization of the product classes in question in these countries.

5. It is necessary to substantiate the originality and relevance of this manuscript more.

6. Authors must be absolutely sure that this manuscript and its separate parts are published for the first time.

7. It is necessary to carefully clarify the significance of the identified differences in tables 4 - 10.

8. If the authors did not use in the discussion any of their previously published works that are close to the subject of this manuscript, then this must be done.

9. It is necessary to provide more detailed biological and chemical information about the investigated products and substances.

Round 2

Reviewer 1 Report (Previous Reviewer 2)

Dear authors,

after corrections and explanations, the manuscript is suitable for publication in Horticulture.

Author Response

Reviewer 4 Report (New Reviewer)

According to the authors' response, this manuscript has been revised for the fourth time.

Certainly, the efforts of the authors could deserve recognition.

However, I still believe, and this is very important both for the manuscript quality and for the target journal, that there is still a need and there is still a potential to strengthen the analytical part of the work.

It is necessary to provide a more conceptual and systemic analysis of the research issues, the results of the work and their discussion.

Thus, the text of the manuscript should be improved as a whole.

As for more specific questions:

  - convert ppm in line 129?

  - when the authors mention alternatives (line 382), it is necessary to clarify - alternatives to what and how is it implemented more specifically?

- it is necessary to more fundamentally substantiate the solution of the task set by the authors to reduce dependence on the use of chemical fertilizers.

- comments and remarks within the framework of the previous round of peer review were not taken into account and covered to the necessary and sufficient extent.

Author Response

This manuscript is a resubmission of an earlier submission. The following is a list of the peer review reports and author responses from that submission.

Round 1

Reviewer 1 Report

There are many improvements to be made in the manuscript (see the attachment for details).

A severe review of the English language are necessary.

The authors discuss the results and effects of biostimulants alone, without taking into account doses and interaction between them.

The introduction and discussion need further discussion and are very superficial.

Journal standards must be reviewed again, for example, the format of references is not standardized and you need to add the doi.

Thus, the manuscript was not yet ready for acceptance by the magazine in its present form.

Reviewer 2 Report

Dear authors, 

The aim of the work was to investigate the influence of different concentrations of humic acids, brassinolide, and seaweed extracts on different parameters of the apricot cultivar Canino for a period of two years. This is an interesting topic considering that the mentioned substances can be a substitute for chemicals that we use in fruit production and have a harmful effect on the environment.

Extensive editing of English language and style required. The manuscript is not clear because of bad English translations and very long sentences. The manuscript is relevant to the field and presented in a well-structured manner.

The aim of the work is not well formulated.

A large number of references are cited, both in the introduction and in the discussion. In addition, the references in the introduction are too general. It is necessary to cite previous research and specific results of the application of the investigated preparations in other fruit species.

The manuscript is not scientifically sound because all paragraphs within the results begin the same way (for example, 3.1. Data in Table 2 showed that foliar application of SWE, Brs, and HA positively improved....; 3.2. Data in Table 3 showed that.... etc.).

An experimental design is not appropriate to test the hypothesis. It is not clearly written how many trees were in one repetition. In the tables, in addition to the mean values, it is necessary to indicate the standard error.

There are many errors in the methodology, for example, it is not stated how the yield was measured, how many leaves were taken to determine the chlorophyll content, and the formula for the percentage of fruit set is not complete.

The conclusion is not well written, it presents only a simple summarization of the research results. A recommendation should be given for the use of the tested preparations based on the obtained results.

Round 2

Reviewer 2 Report

Dear authors,

The manuscript requires reorganization to have a systematic flow and ultimately to improve the clear understanding of the scientific message embedded in it from a horticultural point of view. The discussion and conclusion of the results are very weak. English grammar and expression are also very poor despite the corrections.